# OpenReview forum: "Enhancing LLM Steering through Sparse Autoencoder-Based Vector Refinement"
_ICLR.cc/2026/Conference — ICLR 2026 Conference Withdrawn Submission_

### Official Review · Reviewer_GSBX · 2025-11-01

**Soundness:** 2
**Presentation:** 4
**Contribution:** 3
**Rating:** 2
**Confidence:** 3

**Summary:**

This paper presents a method for enhancing difference-in-means steering by subtracting out non-useful SAE features and adding in useful SAE features. The paper is well written and clearly explains the methods and the experiments. The idea is intuitive and interesting, and proposes to build on a strong baseline steering method of difference-in-means using a new popular method of SAEs. I generally think this paper would be worth accepting, however, there are questions about the evaluations that I need answered before recommending acceptance.

Some missing citations that seem glaring: https://arxiv.org/pdf/2310.06824, https://arxiv.org/abs/2308.10248, https://arxiv.org/abs/2205.05124

The experiments used to evaluate the methods are reasonable. However, a paper that presents a new method like this needs to contextualize results with baselines, especially state-of-the-art baselines, in a fair apples-to-apples comparison. I have concerns about the following:

- I'm concerned that you only did hyperparameter tuning for your SAE-RSV method. The way it is written currently, I'm not sure how you chose the steering factor for other methods or if there even is a steering factor for other methods. If not, I definitely need to see new results where you tune an alpha for each of your baseline methods. This is by far my biggest concern.

- I think you should include a representation fine-tuning baseline. This method is something like an ultra lightweight LoRA or a steering module. I think it's important to see whether how this method does, as it outperforms your other baselines in this benchmark paper: https://arxiv.org/abs/2501.17148.

- I'm concerned that you didn't do any hyperparameter tuning on your LoRA. It's important to put equal effort into making baselines work on a task to the effort you put into making your method work on a task.

- Generally, I'm suspicious of steering evaluations that only have an LM judge look at steering success. I think that having an LM judge for fluency/coherence judgements would be better than using entropy. Sometimes the SAE-RSV is higher entropy than other methods, and I'm not sure how much to take that seriously, i.e., which entropy drops and increases actually manifest in a "bad steer".  Without contrasting pressures between steering and coherence, weird things can happen when you tune hyperparameters only on the steering objective.

I'm willing to increase my score to a weak accept if you clear up my issues with hyperparameter tuning on CAA.

I would increase it more if you address other issues, e.g., tuning the LoRA baseline or training a ReFT baseline.

**Strengths:**

See summary

**Weaknesses:**

See summary

**Questions:**

See summary

---

### Official Review · Reviewer_9pUP · 2025-11-01

**Soundness:** 2
**Presentation:** 2
**Contribution:** 1
**Rating:** 2
**Confidence:** 3

**Summary:**

This paper proposed a steering framework that leverages SAEs to refine steering vectors learned from limited training samples. The approach first removes noising features and then recovers task-relevant features that are often missing in small-sample settings.
The paper demonstrated method effectiveness through experiments on five datasets by showing that it outperformed all the baselines including fine-tuning.

**Strengths:**

The proposed SAE-RSV framework showed strong performance with only 50 training samples per task, highlighting its practical value in low-resource settings where task-specific datasets are limited.

The method was rigorously tested against 7 diverse baselines, including ‌SFT, SAE-adapted steering, linear probes, and standard steering methods, providing a robust comparison with existing methods.

The paper provides detailed ablation studies, separately the validation of the Denoising‌ and ‌Augmentation‌ modules. The authors also  thoroughly evaluated the method’s robustness across different hyperparameters and increasing training data sizes (10 to 1000 samples).

**Weaknesses:**

The core claim of 'selecting noising features based on their semantic irrelevance to the target behavior, rather than relying solely on activation statistic' is only an incremental change in my opinion from previous works on denoise steering vectors with SAEs (Zhao et al, Wang et al, as cited in your paper). The reliance on LLM judge to pick up semantically relevant features is also computationally expensive and harder to scale than simply computing activation similarities.

Also, the foundation of your work assumes that SAE picks up monosemantic features that accurately describe the input behaviour (as stated in your related work). However, many recent works have identified weaknesses of SAEs, such as picking up spurious features and not even outperforming probe-based steering (e.g., Wu et al. (https://arxiv.org/abs/2501.17148); Harle et al. (https://arxiv.org/abs/2506.19382); Chanin et al. (https://arxiv.org/abs/2505.11756)).
Therefore, there is no theoretical guarantee that SAEs recover “true” monosemantic, features with the 'correct' meanings. In other words, how could you be sure that the semantic similarity by SAE features are truly accurate? Did you do any empirically validation on that? Would recommend to at least recognise those works in related work/discussion and discuss how the limitations of SAEs may impact the fundamental assumption of your paper.

The paper only evaluated Llama3-8B-IT model and did not demonstrate generalisability across other models, e.g. Gemma models where trained SAEs are available (Gemma-Scope).

Section 3.1 (how to construct steering vectors) is common sense and completely based on past works, so can be moved to appendix or made much shorter to leave space for more experiments.

**Questions:**

Just for clarification:

How is the semantic meaning of each SAE features being measured? (Section 3.2)

How is the textual explanation of each SAE feature being generated? (Section 3.3)

---

### Official Review · Reviewer_n6Gu · 2025-11-01

**Soundness:** 2
**Presentation:** 3
**Contribution:** 3
**Rating:** 4
**Confidence:** 4

**Summary:**

This paper proposes a new method to enhance steering vectors by projecting them into the SAE space, then removing noisy features and adding task-relevant features that have been missed. Experiments are conducted with Llama-3-8B-Instruct on 5 tasks to show that this method for steering (SAE-RSV) outperforms standard steering methods, e.g. CAA, as well as supervised finetuning. The authors consider whether the performance boost is driven by the denoising or augmentation parts, concluding that they are both useful.

**Strengths:**

This paper presents an interesting new method with strong results. The main strengths are as follows:
- **Writing:** The writing is easy to follow. There are a couple of places where missed parts of the methodology, but in general, it was simple to understand the approach.
- **Main results:** The success rates appear to consistently outperform the standard steering method. This might benefit from some statistical testing to show the significance; however, the delta is relatively consistent across the 5 datasets, which is a positive.
- **Comprehensive analysis:** A particular strength is the follow-up, analysis experiments done. The authors consider which factors in their method drive performance (Table 2), the influence of feature count (Table 3) and the optimal hyperparameters (Table 4). These experiments are interesting and make the main result much more convincing. I particularly like the case study and examples shown in Table 5 -- though how are these features selected? Are they cherry-picked?

**Weaknesses:**

These are in order of importance:
- **Generalisation:** It is a fairly standard review comment, but it would be much more convincing if the paper could show results for a different model. Currently, the method just uses Llama-3-8B-Instruct at layer 25. This analysis is fairly limited. Perhaps the authors could repeat the analysis with Gemma or another open model too? In my opinion, showing generalisation beyond one model is necessary for acceptance to a major conference. Given the results in this paper, I expect it to generalise!
- **Task relevant features:** It would be great to understand why task-relevant features are not included in the activations over the original dataset (i.e. are not in the CAA vector by default). Clearly, there is some empirical benefit from running the LLM over the features and picking out some important ones, but why are these not there already? An analysis and a few examples of this in the paper would be nice.
- **How is the CAA parameter chosen:** For the main experiments (Table 1), how do you set the CAA parameter? One experiment which might be nice is considering the Pareto frontier of SR & Entropy under different CAA parameters and different SAE-RSV parameters (though SAE-RSV having 3 params does make this harder).

**Questions:**

- How do you extract the *textual explanation*. I may have missed this.

---

### Official Review · Reviewer_TUzD · 2025-11-04

**Soundness:** 2
**Presentation:** 3
**Contribution:** 2
**Rating:** 4
**Confidence:** 5

**Summary:**

This paper introduces another technique leveraging SAEs to improve steering vector utility, by using an LLM to identify useful and irrelevant SAE features which were found by similarity to a standard difference-of-means steering vector computed on a dataset and filtered by a language model. The goal is to correct the steering vector (which may be noisy due to small dataset size) using the identified SAE features, under the assumption that the SAE feature labels are informative and more robust than the original steering vector. The methods shows moderate gains on a small dataset of safety-related steering tasks.

**Strengths:**

- This is a practical contribution that takes the best of both steering vectors and SAEs.
- The identified noise and useful SAE features are interesting to analyse and seem to match intuitions about the task.
- The paper is overall clearly written and easy to follow.
- Table 2 is nice since my first question was which parts of the proposed improvements to steering vectors contribute the most to performance.

**Weaknesses:**

- The evaluation dataset is insufficient for making claims about the quality of the method. [Tan et al.](https://proceedings.neurips.cc/paper_files/paper/2024/hash/fb3ad59a84799bfb8d700e56d19c231b-Abstract-Conference.html) have previously noted the unreliability of steering vectors in both ID and OOD settings; all the evals here are limited to ~10 concepts and that to ID; all the prompts are directly related to the steering task, so it's hard to say if the results generalise -- will the anti-corrigibility steering work on arbitrary prompts or just break the model behaviour entirely? While safety tasks are important, generally, I am very concerned that the field of steering vectors is engaging in overfitting to the test set when we work with ID settings for such a tiny dataset and use that to design and score new techniques.
- The opening of the paper cites [Wu et al. (2025)](https://arxiv.org/abs/2501.17148), a common large-scale benchmark for steering with publicly available baseline results, so the author is aware of more comprehensive evaluation setups but does not use them. I don't think I can confidently say the method is good given the tiny scale of evaluation.
- There are many good baselines for SAE-based steering which are not mentioned or compared against. For example, the intro cites [Arad et al. (2025)](https://arxiv.org/abs/2505.20063) which presents a strong approach for SAE-feature selection by filtering for output-affecting vectors but it is not included as a baseline. Also see [Chalnev et al. (2024)](https://arxiv.org/abs/2411.02193), [He et al. (2025)](https://arxiv.org/abs/2505.16188) for other SAE steering baselines, and [Wu et al. (2025b)](https://arxiv.org/abs/2505.20809) for the current SoTA representation steering method.
- There are 3 hyperparameters to tune ($\alpha$ weights) and it seems to me that these need to be tuned separately for every new task. This is inelegant; it would be nice if these could be computed beforehand (e.g. by using the activation differences on the positive and negative datasets).

**Questions:**

- For eq. (4), why would we not want features which large magnitudes to contribute more to the computed noise vector? If an irrelavant feature has large magnitude we should try to remove it more than other features right? Let me know if I don't have the correct intuition here.

---

### Note · Authors · 2025-11-27

I have read and agree with the venue's withdrawal policy on behalf of myself and my co-authors.